# CRAG: Can 3D Generative Models Help 3D Assembly?

**Zeyu Jiang** [*]  **Sihang Li** [*]  **Siqi Tan** [†]  **Chenyang Xu** [†]  **Juexiao Zhang**  **Julia Galway-Witham**  **Xue Wang**
**Scott A. Williams**  **Radu Iovita**  **Chen Feng** [✉]  **Jing Zhang** [✉]

New York University

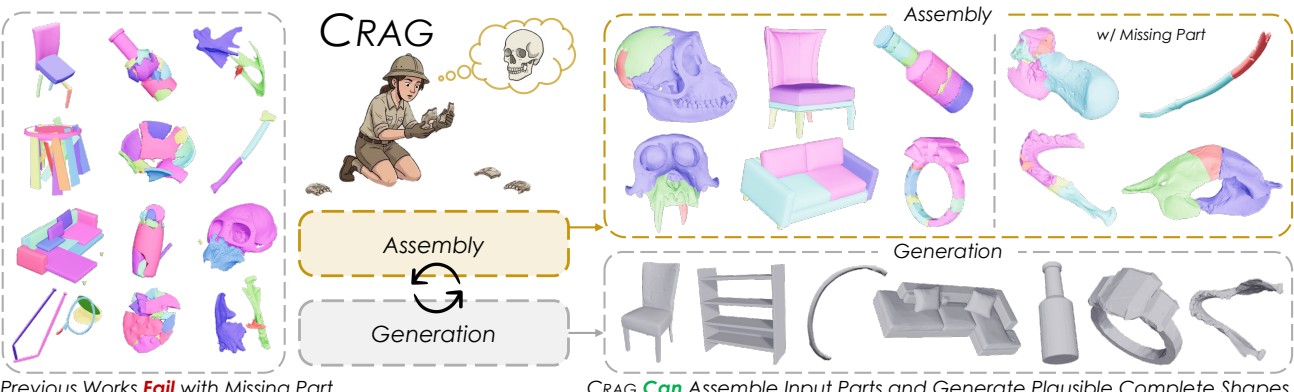

Figure 1: We propose CRAG, a unified framework that couples 3D assembly and generation. CRAG jointly denoises fragment poses and whole-shape latents to assemble the input parts while synthesizing a plausible complete shape, remaining robust to missing parts.

## Abstract

Most existing 3D assembly methods treat the problem as pure pose estimation, rearranging observed parts via rigid transformations. In contrast, human assembly naturally couples structural reasoning with holistic shape inference. Inspired by this intuition, we reformulate 3D assembly as a joint problem of assembly and generation. We show that these two processes are mutually reinforcing: assembly provides part-level structural priors for generation, while generation injects holistic shape context that resolves ambiguities in assembly. Unlike prior methods that cannot synthesize missing geometry, we propose CRAG, which simultaneously generates plausible complete shapes and predicts poses for input parts. Extensive experiments demonstrate state-of-the-art performance across in-the-wild objects with diverse geometries, varying part counts, and missing pieces. Our code and models will be released.

## 1. Introduction

3D assembly aims to reconstruct a complete 3D object from a set of parts or fractured fragments, with broad impact in scientific collections (Falcucci et al., 2025; Abrams et al., 2025), medicine (Zhao et al., 2023; Ge et al., 2022), and robot manipulation (Liu et al., 2024). Given partial and noisy observations, an algorithm must infer how pieces align while maintaining global object coherence. In real-world settings, the problem is further complicated by missing, eroded, or partially scanned pieces (Lu et al., 2025b).

Most learning-based approaches formulate 3D assembly as pose estimation, predicting a rigid transformation for each input piece. For example, GARF (Li et al., 2025a) and PuzzleFusion++ (Wang et al., 2025b) use generative models to estimate 6-DoF alignments for rigid fragments. Recently, Assembler (Zhao et al., 2025a) and RPF (Sun et al., 2025) reparameterize 3D assembly in Euclidean space by predicting each part's point set in the assembled state, and then recovering per-part rigid transformations via least-squares fitting or Procrustes (Gower, 1975) solved with SVD. Assembler (Zhao et al., 2025a) additionally conditions on an image by injecting pretrained visual features to provide global shape cues. However, these approaches are designed to transform the geometry of the observed parts, so they primarily reposition input points and do not synthesize new geometry to fill missing regions.

*, †Equal contribution.

✉ Corresponding authors: {z.jing, cfeng}@nyu.edu.

*Proceedings of the 43rd International Conference on Machine Learning*, Seoul, South Korea. PMLR 306, 2026. Copyright 2026 by the author(s).

In contrast, human experts do not treat assembly as local alignment alone. They iteratively hypothesize the unseen whole while placing fragments, using a progressively refined global shape hypothesis to resolve ambiguity (Wagner, 2013). This is especially critical when pieces are missing, where conservators often infer absent regions from the available fragments and perform gap-filling to restore a plausible, complete form. Motivated by this perspective, we ask an open question: *could 3D assembly and generation be unified to benefit each other?*

Achieving such a unification raises two core challenges. First, one must construct a shared latent space that can embed an unordered set of variable-size, partial, and noisy fragments into a representation that is simultaneously usable for assembly and compatible with a generative shape prior, so that gradients and uncertainty can flow across the two branches rather than being trapped in mismatched latent spaces. Second, even with a shared representation, one must enable *effective bidirectional information exchange*: fragment evidence should steer what the whole should be, while the imagined whole should in turn disambiguate how fragments should align, without creating unstable feedback loops during joint training and inference.

We address these challenges with CRAG, which Couples ReAssembly and Generation in a *joint flow-matching* framework. Concretely, CRAG performs joint denoising by simultaneously predicting an SE(3) flow for fragment poses and a latent-space flow for whole-shape generation. To establish a common "language" between the two tasks, CRAG reuses the VAE from TripoSG (Li et al., 2025b) as a shared embedding space, mapping variable-size fragment sets into features that live in the same latent space as whole-shape generation and are therefore compatible with a strong generative prior. Building on this shared space, CRAG adopts a Mixture-of-Transformers architecture with two parallel branches and introduces a Joint Adapter at each layer. The key design of the Joint Adapter is the bi-directional attention mechanism to enable mutual refinement: fragment features inform what the whole should be, while the imagined whole guides how fragments should align. This design mirrors human experts, who iteratively reconcile fragment observations with hypotheses of the unseen whole. For stable learning, we employ a two-stage training strategy, learning assembly first and then jointly finetuning both tasks.

Our main contributions are as follows:

- **A New Capability for 3D Assembly.** CRAG jointly assembles fragments and synthesizes plausible complete shapes, remaining robust to missing parts while improving alignment under ambiguity.
- **A New Formulation and Framework.** We recast 3D assembly as a coupled reassembly-and-generation objective, and propose a joint flow-matching framework

that denoises fragment poses in SE(3) and whole-shape latents in a single inference loop.

- **SOTA Results and a New Dataset.** CRAG achieves SOTA performance on part assembly (PartNeXt (Wang et al., 2025a)) and fracture reassembly (Breaking Bad (Sellán et al., 2022)). To facilitate future research, we curate and release a new bone fragment dataset from MorphoSource (Boyer et al., 2016).

## 2. Related Work

**3D Assembly.** 3D assembly is a long-standing problem that mirrors human spatial intelligence: it requires reconciling local cues with a coherent global hypothesis under ambiguity. Most methods formulate it as pose estimation, predicting an SE(3) transform for each piece (Lee et al., 2024). Early methods relied on hand-crafted descriptors that generalize poorly. Learning-based methods such as Jigsaw (Lu et al., 2023) and Combinative Matching (Lee et al., 2025) learn correspondences, improving robustness. Recent approaches including GARF (Li et al., 2025a), PuzzleFusion++ (Wang et al., 2025b), and DiffAssemble (Scarpellini et al., 2024) employ generative models that iteratively denoise poses on SE(3). Another line of work leverages complete-shape priors to further constrain this challenging problem. Classical priors rely on explicit templates, e.g., symmetry (Koutsoudis et al., 2010), while recent methods such as RPF (Sun et al., 2025) and Assembler (Zhao et al., 2025a) predict assembled point sets in Euclidean space and recover rigid transforms via SVD. Assembler further uses a reference image to resolve ambiguity, a capability that CRAG also supports through an optional image condition. Jigsaw++ (Lu et al., 2025a) takes a partially assembled point cloud and applies a learned complete-shape prior, but this separated design limits mutual refinement. In contrast, CRAG takes a first step toward jointly coupling assembly and generation within a unified optimization loop, enabling bidirectional benefits.

**3D Generation for Assembly.** Recent advances in 3D generation provide powerful shape priors that can benefit 3D assembly. These models are typically built upon two-stage designs that learn a compact VAE latent space and then model it with diffusion or flow-based transformers (Wu et al., 2024a), often using efficient VecSet latents (Zhang et al., 2023). Representative open models such as Dora (Chen et al., 2025), TripoSG (Li et al., 2025b), and Hunyuan3D 2.0 (Zhao et al., 2025b) follow this paradigm and achieve strong image-to-3D performance by conditioning the latent denoiser on image features. In parallel, TRELLIS (Xiang et al., 2025) introduces a unified structured latent that can be decoded into multiple 3D formats, such as 3D Gaussians and meshes. Beyond image or language prompts, Hunyuan3D-Omni (Hunyuan3D et al., 2025) augments conditioning with point clouds, voxels, 3D bounding boxes, and

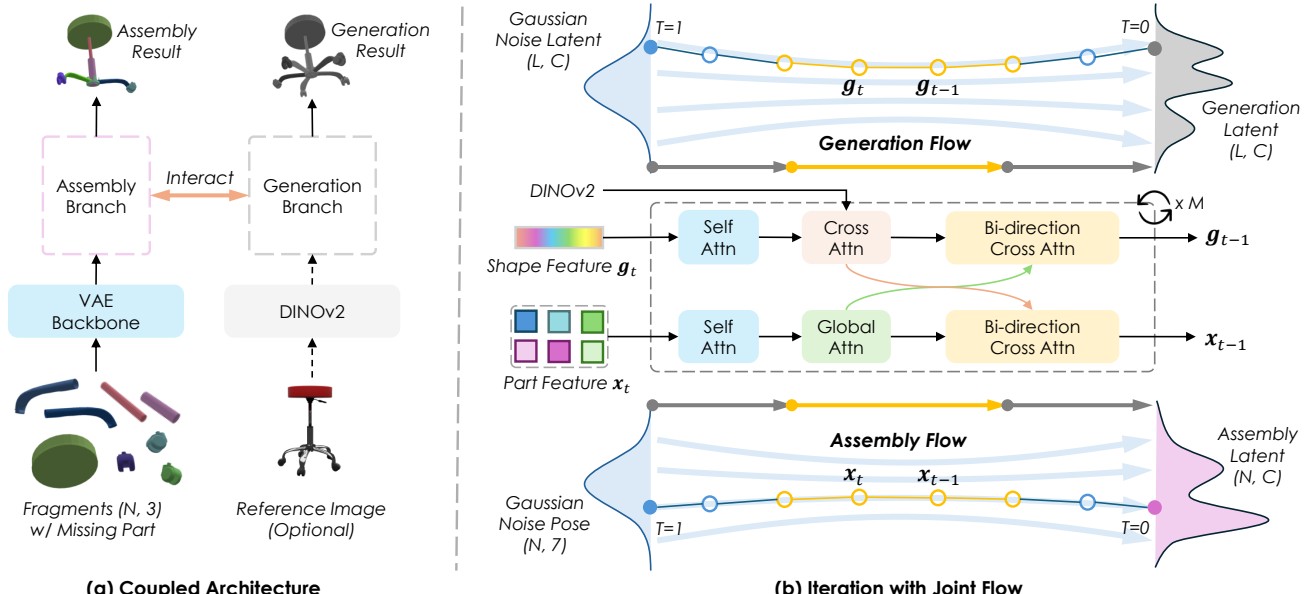

*Figure 2.* Overview of our approach CRAG . We propose a unified framework for 3D assembly and whole-shape generation. Our model consists of two interacting branches: an *Assembly Branch* that predicts the pose for each part via SE(3) flow matching, and a *Generation Branch* that synthesizes the complete shape via flow matching. A *Joint Adapter* bridges these branches, enabling bidirectional information flow. We employ a two-stage training strategy: learning assembly first, and then jointly finetuning both tasks.

skeletal pose priors for fine-grained controllability. CRAG reuses TripoSG's Transformer VAE as a shared VecSet latent space and pre-trained weights as the generation branch, while our joint flow matching allows assembled parts to provide additional structural priors when image cues are ambiguous.

## 3. Method

Given a set of fragments $\mathcal{P} = \{p_i\}_{i=1}^N$ and an optional reference image $I$, our goal is to simultaneously recover the pose $\boldsymbol{T}_i \in \text{SE}(3)$ for each fragment and reconstruct the complete 3D shape $\mathcal{S}$. To achieve this, we propose a unified framework CRAG, as illustrated in Figure 2. In the following subsections, we first review the pre-trained generative backbone, TripoSG (Li et al., 2025b) (§3.1), then describe how we leverage its VAE for fragment embedding (§3.2), and finally introduce our joint flow architecture for pose estimation and shape generation (§3.3).

### 3.1. Preliminary: TripoSG

We first briefly review TripoSG (Li et al., 2025b), the foundational image-to-3D model for our method. TripoSG is a flow-matching-based generative model that operates on a latent set $z \in \mathbb{R}^{n \times d}$, consisting of $n$ latent vectors of dimension $d$. The latent representation is encoded by a transformer-based VAE following the strategy introduced in 3DShape2VecSet (Zhang et al., 2023). Given a 3D shape $\mathcal{S}$, TripoSG first samples $M$ surface points $P \in \mathbb{R}^{M \times 3}$ from $\mathcal{S}$, and then draws a subsampled point set $P' \subset P$. Both point

sets are embedded using a Fourier feature embedding $\gamma(\cdot)$. The initial latent set $z$ is computed by stacking multiple multi-head self-attention (SA) layers on top of a multi-head cross-attention module:

$$z_0 = \text{CrossAttn}\big(\gamma(P'), \gamma(P)\big), \tag{1}$$

$$z = \text{Linear}\big(\text{SelfAttn}_i(z_0)\big), \quad i \in [1, L_{\text{SA}}^E], \tag{2}$$

where $L_{\text{SA}}^E$ denotes the number of SA layers in the encoder. The output $z$ serves as the noise-adding target for the denoiser model. Afterwards, $z$ can be decoded by another set of self-attention layers, followed by a cross-attention to decode the signed distance function (SDF) values at the query points $x$.

$$\tilde{z} = \text{Linear}\big(\text{SelfAttn}_j(z)\big), \quad j \in [1, L_{\text{SA}}^D], \tag{3}$$

$$\tilde{s} = \text{CrossAttn}\big(\gamma(x), \tilde{z}\big), \tag{4}$$

where $L_{\text{SA}}^D$ denotes the number of SA layers in the decoder.

### 3.2. Shared VAE for Fragment Embedding

**Why Using Shared VAE.** Most recent fragment assembly methods learn fragment-level representations by solving task-specific binary classification problems, such as predicting fracture surfaces (Li et al., 2025a) or detecting fragment overlaps (Sun et al., 2025). However, such encoders are inherently limited by the fragmentation patterns present in their training data and often fail to generalize to unseen fragmentations. In contrast, we propose to reuse the VAE

from TripoSG (Li et al., 2025b) as a *shared fragment embedding module*. Since this VAE is trained on large-scale 3D shape datasets, it provides a more robust and generalizable geometric prior. More importantly, sharing the same VAE between part assembly and whole-shape generation establishes a unified latent space, which facilitates effective information exchange between the two branches.

**Fragment Encoding.** To support variable numbers of fragments and uneven fragment sizes, we first adapt the VAE encoder with a customized attention processor that allows variable-length input. Given a set of fragments $\mathcal{P} = \{p_i\}_{i=1}^N$, we sample $M_i$ points proportional to each fragment's area, then downsample by a factor of 4 to form the VAE query set $P_i'$. Each fragment is then independently encoded using the shared VAE encoder, producing a fragment-specific latent feature. Instead of directly using encoded latent $z$ as the fragment representation, we use the decoded latent $\tilde{z}$ as the fragment feature. We found that this choice provides better alignment between the assembly and generation branches and yields better performance.

### 3.3. Joint Flow Matching

The proposed joint flow matching adopts a Mixture-of-Transformers architecture that coordinates the assembly and generation processes. As shown in Figure 2, the model consists of two parallel Transformer branches: an *Assembly Branch* that predicts the per-fragment SE(3) flow, and a *Generation Branch* that predicts the flow for the shape latents. A *Joint Adapter* serves as a bridge, enabling bidirectional information flow between the two branches at each layer.

**Assembly Branch.** Following GARF (Li et al., 2025a), we model the assembly process as a continuous flow on the manifold $\mathcal{M} = \mathrm{SO}(3) \times \mathbb{R}^3$. Given an initial state $(\boldsymbol{r}_0, \boldsymbol{a}_0)$ and a target state $(\boldsymbol{r}_1, \boldsymbol{a}_1)$, the flow trajectory is defined as:

$$\begin{aligned} \boldsymbol{r}_t &= \exp_{\boldsymbol{r}_0}\left(t \log_{\boldsymbol{r}_0}(\boldsymbol{r}_1)\right), \\ \boldsymbol{a}_t &= (1-t)\boldsymbol{a}_0 + t\boldsymbol{a}_1, \end{aligned} \quad (5)$$

where $t \in [0, 1]$. Here, $\boldsymbol{r}$ represents orientation (handled via geodesics on SO(3)) and $\boldsymbol{a}$ represents position (linear interpolation). We train the flow matching network by minimizing the difference between the predicted velocity and the ground-truth vector field:

$$\begin{aligned} \mathcal{L} = \mathbb{E}_{t,(\boldsymbol{r}_1,\boldsymbol{a}_1)}\Bigg[ \sum_{i=1}^N \Big( &\left\| f_r^i(\boldsymbol{r}_t, \boldsymbol{a}_t, t) - u_t \right\|^2 \\ &+ \left\| f_a^i(\boldsymbol{r}_t, \boldsymbol{a}_t, t) - v_t \right\|^2 \Big) \Bigg], \end{aligned} \quad (6)$$

where the flow targets are defined as $u_t = \frac{\log_{\boldsymbol{r}_t}(\boldsymbol{r}_1)}{1-t}$ and $v_t = \frac{\boldsymbol{a}_1 - \boldsymbol{a}_t}{1-t}$. We omit the fragment index $i$ for brevity.

**Generation Branch.** While the assembly branch operates on fragment-level poses, the generation branch works in the latent space to reconstruct the complete shape. Following TripoSG (Li et al., 2025b), we model the generation process as a continuous-time flow between the clean latent $z_0 \in \mathbb{R}^{n \times d}$ (encoded by the shared VAE) and Gaussian noise $z_1 \sim \mathcal{N}(0, I)$. At each time step $t \in [0, 1]$, the generation transformer predicts the velocity field:

$$v_z = f_{\mathrm{gen}}(z_t, t, c_I), \quad (7)$$

where $c_I$ is an optional image condition extracted from the reference image $I$ using a frozen DINOv2 (Oquab et al., 2023) encoder. When no reference image is provided, $c_I$ is set to zero. The training objective minimizes MSE between predicted and ground-truth velocities $\mathcal{L}_{\mathrm{gen}} = \mathbb{E}[\|v_z - (z_0 - z_1)\|^2]$. During inference, we iteratively denoise from $z_1$ to $z_0$ using the Euler method, and decode the final shape $\mathcal{S}$ using the shared VAE decoder.

Importantly, the generation transformer is initialized from the pre-trained TripoSG (Li et al., 2025b) model and can be either frozen or fine-tuned, allowing us to leverage powerful geometric priors from large-scale 3D datasets while optionally adapting to fragmented objects.

**Joint Adapter.** A key challenge in our unified framework is enabling effective information exchange between the assembly and generation branches. While both branches condition on the global image context $c_I$, they operate on fundamentally different representations: the assembly branch processes variable-length fragment point clouds, whereas the generation branch works with fixed-size latent tokens. To bridge this gap, we introduce a *Joint Adapter* module at each transformer layer.

Let $h_{\mathrm{asm}}^{(\ell)} \in \mathbb{R}^{M \times d}$ and $h_{\mathrm{gen}}^{(\ell)} \in \mathbb{R}^{L \times d}$ denote the hidden states of the assembly and generation branches at layer $\ell$, respectively. The Joint Adapter facilitates bidirectional information flow through a symmetric cross-attention mechanism. Formally, the hidden states are updated as:

$$\begin{aligned} h_{\mathrm{gen}}^{(\ell)} &= h_{\mathrm{gen}}^{(\ell)} + \mathrm{CrossAttn}\big(h_{\mathrm{gen}}^{(\ell)}, h_{\mathrm{asm}}^{(\ell)}\big), \\ h_{\mathrm{asm}}^{(\ell)} &= h_{\mathrm{asm}}^{(\ell)} + \mathrm{CrossAttn}\big(h_{\mathrm{asm}}^{(\ell)}, h_{\mathrm{gen}}^{(\ell)}\big), \end{aligned} \quad (8)$$

where $\mathrm{CrossAttn}(x, y)$ computes the multi-head attention with $x$ serving as the query and $y$ as the key/value context. This formulation allows the generation branch to query geometric details from the fragments, while the assembly branch simultaneously incorporates structural priors from the generation latent space. Crucially, to ensure stable training and preserve the pre-trained priors of the generation model, we initialize the output projection layers of the adapter with zero weights. This ensures that the adapter acts as an identity mapping at the beginning of training.

*Table 1.* Quantitative comparison on PartNeXt (Wang et al., 2025a) and Breaking Bad (Sellán et al., 2022) under two part-status settings: *Complete* (all parts observed) and *Missing* (with missing parts). CRAG consistently achieves the best overall performance across datasets and remains robust in the challenging missing-part setting. The **best** and second best results are highlighted.

| | PartNeXt (Wang et al., 2025a) | | | | | | | | Breaking Bad (Sellán et al., 2022) | | | | | | | |
|---|---|---|---|---|---|---|---|---|---|---|---|---|---|---|---|---|
| Part Status | Complete | | | | Missing | | | | Complete | | | | Missing | | | |
| Method | RE↓ | TE↓ | PA↑ | CD↓ | RE↓ | TE↓ | PA↑ | CD↓ | RE↓ | TE↓ | PA↑ | CD↓ | RE↓ | TE↓ | PA↑ | CD↓ |
| GARF (Li et al., 2025a) | 52.52 | 10.68 | 58.19 | 3.10 | 49.40 | 9.73 | 60.71 | 5.78 | 8.75 | 1.72 | 93.56 | 0.42 | 14.59 | 3.01 | 85.55 | 3.43 |
| RPF (Sun et al., 2025) | 54.99 | 29.54 | 46.17 | 10.46 | 42.49 | 22.95 | 57.20 | 8.21 | 30.59 | 13.67 | 80.23 | 1.01 | 31.04 | 14.96 | 77.05 | 1.20 |
| CRAG w/o img (Ours) | 45.45 | 9.82 | 61.67 | 3.31 | 42.46 | 8.70 | 66.74 | 5.17 | **8.00** | 1.37 | 94.64 | 0.22 | **11.43** | 1.79 | **92.03** | 0.52 |
| Assembler (Zhao et al., 2025a) | 85.82 | 17.14 | 44.18 | 27.93 | 83.80 | 15.00 | 49.86 | 24.71 | 74.92 | 13.22 | 48.38 | 7.46 | 74.45 | 13.43 | 45.93 | 7.78 |
| CRAG (Ours) | **45.12** | **9.13** | **65.89** | **2.40** | **42.33** | **7.86** | **71.81** | **4.21** | **8.00** | **1.36** | **94.68** | **0.21** | 11.44 | **1.77** | 92.07 | **0.50** |

# 4. Experiment

## 4.1. Implementation Details

CRAG is built upon the pretrained TripoSG (Li et al., 2025b) and adopts its pretrained VAE for part-level feature extraction. The assembly branch follows GARF's (Li et al., 2025a) design, but extends to 21 layers to match the depth of the pretrained generation branch with skip connections. To stabilize training and accelerate convergence, we employ a two-stage training strategy. In the first stage, only the assembly branch is trained for 100k steps for warm-up. In the second stage, the generation branch is activated with joint adapters, and the entire model is jointly trained for an additional 150k steps. We keep the classifier-free guidance strategy in the second stage, but increase the image condition drop rate to 50% from 10% along the training process to enable the generation branch to learn from the part-level assembly information. The full training process takes about 3 days on 32 NVIDIA H200 GPUs, with a global batch size of 256 in the first stage and 128 in the second stage.

## 4.2. Experiment Setup

**Datasets.** We define two tasks, *part assembly* and *fracture reassembly*, and train and evaluate our method on each task separately. (i) *Part Assembly*. We use **PartNeXt** (Wang et al., 2025a) as both the training dataset and evaluation benchmark. PartNeXt contains 23,519 textured 3D models, from which we sample 15,563 shapes for training and 3,903 shapes for evaluation, restricting the number of parts per shape to the range of 2 to 20. PartNeXt focuses on *semantic parts* (e.g., chair legs and table tops), which are defined by functional or semantic boundaries rather than geometric fracture surfaces. (ii) *Fracture Reassembly*. For this task, we use the everyday subset of the **Breaking Bad** (Sellán et al., 2022) dataset, combined with a curated collection from **MorphoSource** (Boyer et al., 2016). MorphoSource is an open-access repository containing thousands of 3D models of primate (including human) bones and bones from other animals. We virtually fracture these models using Breaking Good (Sellán et al., 2023), FractureRB (Hahn & Wojtan, 2016), and FractureBEM (Hahn & Wojtan, 2015) to

substantially increase sample size and taxonomic coverage. The resulting dataset contains 3,347 samples across 10 categories. Complementarily, we use the **FRACTURA** dataset (Li et al., 2025a) to validate CRAG on real-world fractures.

**Evaluation Metrics.** Following (Li et al., 2025a; Sun et al., 2025; Wang et al., 2025b), we evaluate the assembly quality by following metrics: (i) **Rotation Error (RE)** is the root mean square error of Euler angles for each part. (ii) **Translation Error (TE)** measures the root mean square error of translation vectors for each part. (iii) **Part Accuracy (PA)** is the fraction of parts whose the chamfer distance to the ground truth is less than $10^{-2}$. (iv) **Chamfer Distance (CD)** computes the chamfer distance between the assembled shape and the ground truth shape. All metrics are averaged over all parts and all shapes.

**Baseline Methods.** We compare our approach against state-of-the-art methods for 3D part assembly. **GARF** (Li et al., 2025a) and **RPF** (Sun et al., 2025) are point cloud-based methods that take only part geometry as input. GARF directly regresses the SE(3) pose for each part, while RPF first predicts the target location of each part's point cloud and subsequently recovers the rigid transformation through SVD optimization. **Assembler** (Zhao et al., 2025a) is the most related work to ours, which augments the input with multi-view images to provide additional visual context. Its overall pipeline follows a similar predict-then-optimize paradigm as RPF. We denote our method as **CRAG** and report results on both datasets described above. Additionally, we extend the evaluation protocol to include scenarios with *missing parts* during inference.

## 4.3. Evaluation

We conduct extensive experiments to address three central questions of this work:

**Q1:** Does generation improve assembly by providing a holistic prior?

**Q2:** Can we still assemble the observed parts and synthesize a complete object from incomplete fragment sets?

**Q3:** Does part-level evidence help disambiguate image-conditioned 3D generation?

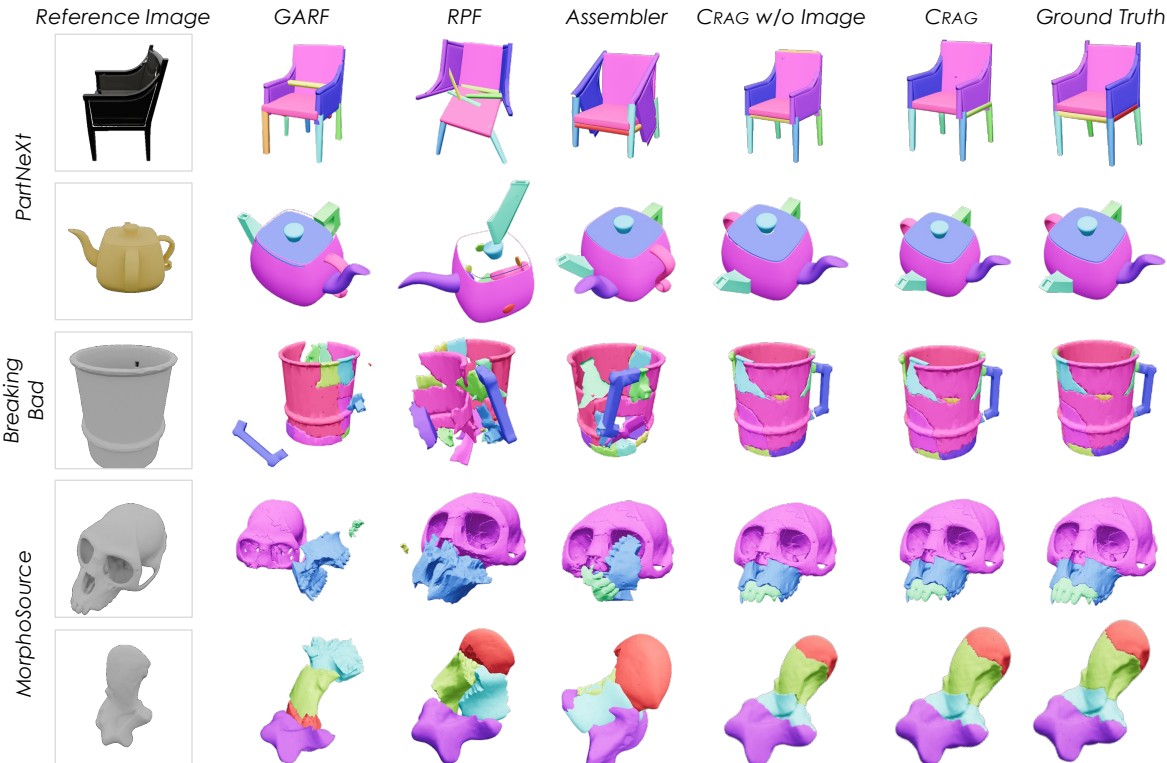

*Figure 3.* Qualitative results across PartNeXt (Wang et al., 2025a), Breaking Bad (Sellán et al., 2022), and MorphoSource (Boyer et al., 2016). We first compare methods without reference images by contrasting GARF (Li et al., 2025a), RPF (Sun et al., 2025), and CRAG w/o image, where CRAG produces more coherent assemblies and more complete shapes from the same observed parts. We then compare image-conditioned methods by showing Assembler and full CRAG given the reference image, where CRAG better aligns parts and yields shapes that more closely match the ground truth.

**Holistic Shape Priors Boost Assembly Performance.** Table 1 reports quantitative results on PartNeXt (Wang et al., 2025a) and Breaking Bad (Sellán et al., 2022) under the complete-part setting. We evaluate two variants of our approach: **CRAG**, which uses a reference image, and **CRAG w/o img**, which uses only part point clouds. To answer **Q1** fairly, we first compare methods without reference images, where CRAG w/o img consistently outperforms GARF (Li et al., 2025a) and RPF (Sun et al., 2025) across datasets, indicating that coupling assembly with generation provides a holistic prior that regularizes pose recovery from geometry alone. When reference images are available, CRAG further surpasses the image-conditioned Assembler (Zhao et al., 2025a) by a large margin. On PartNeXt, CRAG increases PA from 44.18 to 65.89 and reduces CD from 27.93 to 2.40, corresponding to a 91.4% relative reduction.

These quantitative trends are also evident in Figure 3. GARF (Li et al., 2025a) and RPF (Sun et al., 2025) often produce unstable pose estimates under ambiguity, leading to visibly inconsistent part placements such as tilted or floating structures on PartNeXt (Wang et al., 2025a) and severe misalignments on Breaking Bad (Sellán et al., 2022) and MorphoSource (Boyer et al., 2016). In contrast, CRAG w/o img yields coherent assemblies across all three datasets,

preserving global structure and maintaining consistent part-to-part contacts, which suggests that the coupled generation prior provides a strong holistic constraint beyond local geometric cues. When conditioned on the reference image, Assembler (Zhao et al., 2025a) can reduce some ambiguities but still exhibits implausible global structure or residual part misplacement, whereas full CRAG produces assemblies that more closely match the ground truth and simultaneously synthesizes more complete and globally consistent shapes. Overall, these results answer **Q1** affirmatively:

> **A1:** *The generation prior provides holistic structural guidance that improves assembly.*

**Robust Assembly and Shape Synthesis with Missing Parts.** In the missing-part setting of Table 1, CRAG remains robust even without reference images. CRAG w/o img consistently outperforms GARF (Li et al., 2025a) and RPF (Sun et al., 2025) on both datasets, with higher part accuracy and lower shape error, showing stronger assembly performances from incomplete observations. On PartNeXt (Wang et al., 2025a), it increases PA to 66.74 while reducing CD to 5.17, improving over GARF and RPF. On Breaking Bad (Sellán

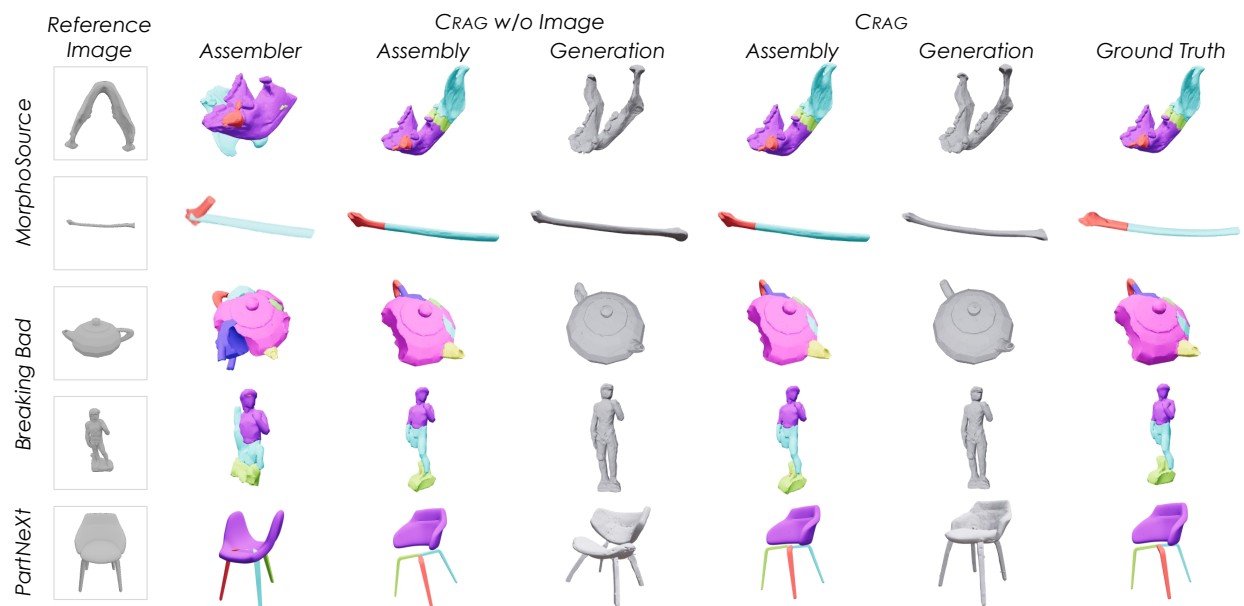

*Figure 4.* Qualitative results on PartNeXt (Wang et al., 2025a), Breaking Bad (Sellán et al., 2022), and MorphoSource (Boyer et al., 2016) with missing parts. We compare Assembler, CRAG without reference images, and CRAG given a reference image. CRAG simultaneously assembles the observed parts and synthesizes a plausible, complete shape, and reference images further improve fidelity when available.

et al., 2022), the gap is larger, reaching 92.03 PA with 0.52 CD, substantially surpassing both baselines under severe incompleteness. When reference images are available, CRAG further improves and clearly exceeds the image-conditioned Assembler (Zhao et al., 2025a), which degrades sharply with missing parts. Figure 4 visualizes CRAG's joint assembly-and-generation behavior: it not only places the available parts coherently but also synthesizes a globally consistent complete shape by hallucinating the missing geometry, with reference images further improving the fidelity of the generated completion when available.

**Comparison with Shape-Completion Baselines.** A natural alternative to our joint formulation is a sequential pipeline: first assemble the fragments, then apply a shape-completion model on the assembled result as a downstream step to fill in the missing geometry. To examine this alternative, we compare against two representative image-conditioned shape-completion methods, SDFusion (Cheng et al., 2023) and MGPC (Liu et al., 2026), under the same PartNeXt (Wang et al., 2025a) missing-part setting (2–20 parts). We grant these baselines an oracle upstream stage: they receive fragments already pre-aligned at the ground-truth poses, treated as a single point cloud with missing regions, together with a reference image. CRAG, in contrast, starts from multiple *unaligned* fragments and must solve assembly and shape synthesis jointly. As shown in Table 2, CRAG substantially outperforms both baselines across all metrics despite their oracle assembly; even CRAG w/o image, which has neither reference image nor pre-aligned fragments, surpasses image-guided MGPC, showing that jointly reasoning about

assembly and shape generation is more informative than a sequential assembly-then-complete pipeline even when the latter is handed the assembly for free.

Together, these results answer **Q2**:

> **A2:** CRAG *can robustly assemble the observed parts and simultaneously synthesize a plausible complete object, even when parts are missing.*

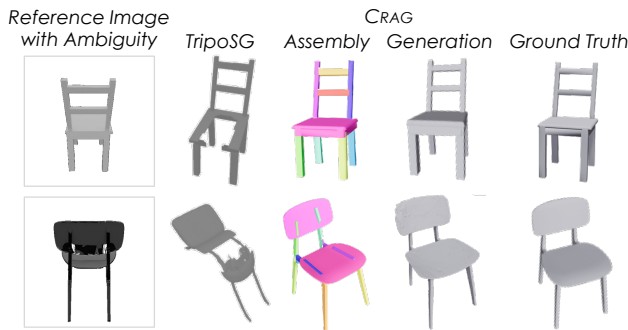

*Figure 5.* Qualitative results under ambiguous reference images on PartNeXt (Wang et al., 2025a). We compare image-only generation with TripoSG (Li et al., 2025b) against CRAG by visualizing CRAG's assembled parts and generated shapes alongside the ground truth. When the reference view is incomplete and does not reveal the full object, part-level evidence helps, to some extent, resolve ambiguity and yields a better shape.

**Part-Level Evidence Helps Disambiguate Image-Conditioned Generation.** We evaluate generation quality on the PartNeXt (Wang et al., 2025a) evaluation set (4,626

*Table 2.* Comparison with shape-completion baselines on PartNeXt (Wang et al., 2025a) under the missing-part setting (2–20 parts). SDFusion (Cheng et al., 2023) and MGPC (Liu et al., 2026) operate as downstream completion modules: they receive fragments already pre-aligned at ground-truth poses (equivalent to an oracle upstream assembly) as a single point cloud with missing regions, together with a reference image. CRAG starts from unaligned fragments and must solve both assembly and shape synthesis jointly, yet still substantially outperforms both.

| Method | CD-L1 ($\times 10^{-2}$)↓ | CD-L2 ($\times 10^{-3}$)↓ | F-Score@1% (%)↑ | F-Score@5% (%)↑ | EMD ($\times 10^{-3}$)↓ |
|---|---|---|---|---|---|
| SDFusion (Cheng et al., 2023) | 22.51 | 23.93 | 1.90 | 28.58 | 28.60 |
| MGPC (Liu et al., 2026) | 12.59 | 11.54 | 12.75 | 70.54 | 16.99 |
| CRAG w/o img | 12.31 | 10.32 | 13.84 | 74.11 | 16.12 |
| CRAG | **9.39** | **5.76** | **18.23** | **82.82** | **13.09** |

shapes), where each object is pre-rendered from multiple viewpoints and one view is randomly sampled as the reference image at evaluation time. When the sampled view happens to be uninformative, heavily occluded or failing to reveal the full object, image-only TripoSG (Li et al., 2025b) either produces geometrically inconsistent reconstructions or fails to decode altogether: as reported in Table 3, TripoSG fails entirely on 25.0% (1,157/4,626) of inputs, while CRAG w/o img, which uses no reference image at all, still produces a valid shape and attains comparable generation quality from fragment evidence alone. Combining the two, full CRAG yields consistent gains across all metrics over image-only TripoSG, confirming that fragment geometry provides structural constraints that complement image cues. Figure 5 illustrates this qualitatively: when the reference view does not reveal the full object, TripoSG produces inconsistent shapes, while CRAG uses the assembled parts as additional evidence to generate shapes closer to the ground-truth. These observations answer **Q3**:

> **A3:** *To some extent, part-level evidence can reduce ambiguity in image-conditioned 3D generation by providing additional geometric constraints when the reference view is incomplete.*

**Ablation Study.** Table 4 ablates key design choices in CRAG on PartNeXt (Wang et al., 2025a). **(i) Fragment Encoding.** ① uses a PTv3 (Wu et al., 2024b) encoder with GARF-style task-specific pretraining (e.g., fracture-surface segmentation), while ② replaces it with our adapted TripoSG VAE embedding (decoded $\tilde{z}$). The consistent gains in ② suggest that large-scale 3D generative pretraining provides a stronger geometric prior for assembly. **(ii) Generation Branch.** To quantify how much the generation branch helps 3D assembly beyond simply adding an image condition, we compare assembly-only with images ③ against the full coupled model ⑤. The gain from ③ to ⑤ indicates that coupling assembly with generation provides additional

holistic shape context that resolves ambiguities not captured by image features alone. **(iii) Robustness without Reference Images.** Comparing ④ against ⑤, together with the strong performance of ④, shows that CRAG remains effective even without a reference image: the generative prior can still regularize and guide pose denoising, while images further boost performance when available.

**Robustness Scales with Fragment Count.** To examine how assembly difficulty scales with the number of parts, we break down PA on PartNeXt (Complete) by the number of parts per shape (2–10) in Table 5. Both methods degrade as the number of parts grows, as more parts induce greater combinatorial ambiguity, but CRAG's lead over Assembler (Zhao et al., 2025a) widens substantially, roughly doubling from $+11.4$ PA at 2 parts to $+22.8$ PA at 10 parts. This also indicates that the holistic shape prior contributed by the generation branch is increasingly valuable when assembly cannot be resolved from local part-to-part cues alone.

**Real-World Fractures.** In Figure 6, we visualize CRAG on the FRACTURA (Li et al., 2025a) to validate performance on real-world fractures.

**Representative Failure Cases.** In Figure 7, we observe three representative failure modes of CRAG. First, thin shell fragments provide weak pose constraints, so the assembly flow may converge to a plausible but incorrect SE(3) alignment. Second, the TSDF-based VAE encoding underrepresents slender components when their thickness falls below the truncation distance, causing opposite surfaces to merge and producing broken structures. Third, for non-watertight surfaces, signed distance is ambiguous near boundaries, biasing reconstructions toward watertight shapes with hole closing or sheet thickening, and these errors propagate through the assembly generation coupling.

## 5. Conclusion and Discussion

We present CRAG, a joint flow-matching framework that couples 3D reassembly and whole-shape generation in the wild. CRAG reuses the TripoSG VAE and its large-scale pretrained generative weights to anchor a shared latent space,

---

We count a generation as failed when TripoSG's VAE-decoded output is empty or degenerate under marching-cubes extraction, so no usable mesh is produced.

*Table 3.* Generation quality on the PartNeXt (Wang et al., 2025a) evaluation set (4,626 shapes). Part-level evidence from the assembly branch substantially improves image-conditioned generation across all metrics and prevents the VAE-decoding failures observed in image-only TripoSG (Li et al., 2025b).

| Method | CD-L1 ($\times 10^{-2}$)↓ | CD-L2 ($\times 10^{-3}$)↓ | F-Score@1% (%)↑ | F-Score@5% (%)↑ | EMD ($\times 10^{-3}$)↓ | Failure Rate ↓ |
|---|---|---|---|---|---|---|
| TripoSG (Li et al., 2025b) | 9.12 | 11.23 | 14.63 | 73.18 | 16.67 | 25.0% (1157/4626) |
| CRAG w/o img | 9.26 | 11.46 | 15.02 | 72.61 | 16.96 | **0%** |
| CRAG | **6.53** | **5.83** | **20.49** | **83.87** | **12.92** | **0%** |

*Table 4.* Ablation study (PartNeXt (Wang et al., 2025a), Complete) with key design choices. "A" and "G" refer to the assembly branch and the generation branch, respectively.

| Setups | Enc. | Ref. Img | Branch | RE ↓ | TE ↓ | PA ↑ | CD ↓ |
|---|---|---|---|---|---|---|---|
| ① | PTv3 | × | A | 52.52 | 10.68 | 58.19 | 3.10 |
| ② | VAE | × | A | 47.16 | 10.12 | 60.01 | 3.61 |
| ③ | VAE | ✓ | A | 47.13 | 9.60 | 62.29 | 2.69 |
| ④ | VAE | × | A + G | 45.45 | 9.82 | 61.67 | 3.31 |
| ⑤ (CRAG) | VAE | ✓ | A + G | **45.12** | **9.13** | **65.89** | **2.40** |

*Table 5.* PA on PartNeXt (Wang et al., 2025a) (Complete) broken down by the number of parts per shape. CRAG's lead over Assembler (Zhao et al., 2025a) (Δ = CRAG − Assembler, last row) widens substantially as the number of parts grows, roughly doubling from +11.4 at 2 parts to +22.8 at 10 parts.

| # Parts | 2 | 3 | 4 | 5 | 6 | 7 | 8 | 9 | 10 |
|---|---|---|---|---|---|---|---|---|---|
| Assembler | 83.6 | 76.0 | 64.7 | 54.0 | 44.5 | 38.8 | 33.6 | 34.0 | 31.1 |
| CRAG | **95.0** | **91.2** | **84.7** | **70.9** | **65.6** | **62.0** | **57.0** | **58.9** | **53.9** |
| Δ | +11.4 | +15.2 | +20.0 | +16.9 | +21.1 | +23.2 | +23.4 | +24.9 | +22.8 |

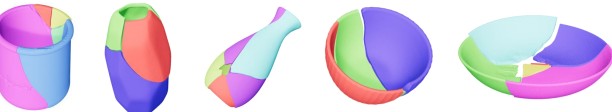

*Figure 6.* Qualitative results of CRAG on FRACTURA (Li et al., 2025a), demonstrating robustness on real-world fractures. All parts are real scanned fragments; colors are rendered for visualization.

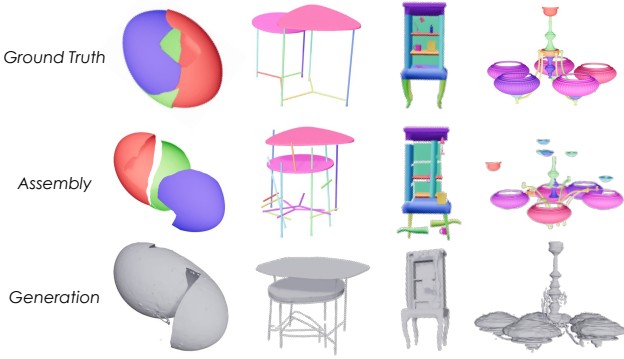

*Figure 7.* Our representative failure cases.

then jointly denoises fragment poses on SE(3) and whole-shape latents, with a Joint Adapter enabling bidirectional information exchange between the two branches. Across extensive benchmarks, we find that this coupling injects a holistic shape context that improves assembly under missing-part settings, while part-level evidence helps disambiguate image-conditioned generation.

**Broader Impacts.** Across archaeology and paleoanthropology, 3D assembly transforms fragmented artifacts, bones, and fossil scans into coherent digital specimens, enabling scalable and reproducible morphometric analysis and cross-site comparison beyond what manual refitting can support. CRAG can further speed hypothesis testing on anatomy and evolutionary variation, particularly for incomplete or eroded specimens. In medicine, analogous multi-fragment reconstruction from CT supports preoperative assessment, surgical planning, and reduction guidance, complementing robot-assisted fracture reduction that seeks higher accuracy and safety. In robotics, everyday repair and maintenance require reasoning about spatial part relations under occlusion and ambiguity, and CRAG highlights how coupling part evidence with global shape hypotheses can improve manipulation when sensory cues are sparse.

**Limitations and Future Work.** First, CRAG is affected by distribution bias in training data. While PartNeXt (Wang et al., 2025a) is large, it overrepresents common categories with canonical part structures, leaving the long tail underrepresented and limiting OOD generalization, which motivates the need for broader community-scale part assembly data. Second, metrics like PA and CD measure geometry but can miss semantic correctness under symmetry and interchangeable parts, such as swapping identical table legs, calling for permutation and symmetry-aware evaluation. Finally, CRAG currently uses an image-conditioned interface inherited from TripoSG (Li et al., 2025b), whereas many applications need richer controls, such as sketches and language, for hypothesis-driven reconstruction and staged objectives. Overall, CRAG shows the promise of joint inference, but improving data coverage, evaluation, and multimodal controllability remains key future work.

## Acknowledgments

This work was supported in part by NSF Grants 2152565, 2238968, and 2514030, and by NYU IT High Performance Computing resources, services, and staff expertise. This research was also supported by the NVIDIA Academic Grant Program using NVIDIA RTX 6000 Ada GPUs.

## Impact Statement

This paper advances machine learning methods for 3D assembly and generative reconstruction, with potential benefits for cultural heritage preservation, archaeological analysis, digital restoration, and robotics. The method may help experts explore plausible assembly and completion hypotheses from fragmented or incomplete 3D observations. However, because generative models can produce plausible but incorrect reconstructions, their outputs should be interpreted as hypotheses rather than definitive conclusions, especially in archaeological or scientific settings. We do not foresee direct negative societal impacts beyond the risk of over-interpretation, which should be mitigated through expert validation and uncertainty-aware analysis.

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

# A. Implementation Details

## A.1. Datasets

We curate a new dataset on the MorphoSource (Boyer et al., 2016) osteological collection. The dataset comprises a diverse range of bone fragments categorized by anatomical regions. We summarize the detailed statistics of the dataset distribution in Table 6.

Table 6. Distribution of osteological samples by anatomical category in the MorphoSource (Boyer et al., 2016) collection.

| Category | Main Anatomical Composition | Count |
|---|---|---|
| Cranium | Cranium elements (cranium, maxilla, frontal, zygomatic) | 1,130 |
| Mandible | Mandible elements (mandible, hemimandible) | 1,015 |
| Forearm | Ulna, radius | 489 |
| Shoulder Girdle | Scapula, clavicle | 198 |
| Leg & Knee | Tibia, fibula, patella, femur fragments | 173 |
| Vertebral Column | Vertebrae (cervical, thoracic, lumbar), sacrum | 129 |
| Pelvis | Pelvis (ilium, ischium, pubis) | 125 |
| Thorax | Ribs, sternum, manubrium | 88 |
| **Total** | | **3,347** |

## A.2. Data Preprocessing

We employ a unified data preprocessing pipeline across all datasets, with task-specific normalization to accommodate different geometric priors.

For the *reassembly* task, each object is represented by a point cloud of $N = 20{,}000$ points sampled from the object surface, where points are distributed among parts proportional to their surface area. Each part is normalized independently to fit within a cube of $[-0.5, 0.5]^3$. This tighter normalization encourages the assembly branch to focus on relative part geometry and pose relationships, rather than absolute scale.

For the *generation* task, we similarly sample $N = 20{,}000$ points uniformly at random from the entire object surface. Point clouds are normalized to a larger bounding volume of $[-0.95, 0.95]^3$ to better preserve global shape proportions and to align with the scale assumptions of the shared VAE. In particular, this normalization is consistent with TripoSG (Li et al., 2025b), which performs Marching Cubes (Lorensen & Cline, 1987) extraction within a cubic volume of size $[1.05]^3$.

All point clouds are centered at the origin. We perform PCA-based (Wold et al., 1987) canonical alignment for each individual part, aligning its principal axes to a canonical coordinate frame. This step is critical for stable training and robust performance, especially when handling geometrically similar or interchangeable components.

For data augmentation, we apply a random global rotation sampled uniformly from $\mathrm{SO}(3)$ to the input object during training for the reassembly task. In contrast, the generation task operates exclusively in the canonical orientation to encourage learning a consistent global shape distribution.

## A.3. Shared VAE for Fragment Embedding

Given an input point cloud represented by coordinates and normals $\boldsymbol{P} \in \mathbb{R}^{N \times 6}$, we first apply Farthest Point Sampling (FPS) *independently for each part* to obtain a reduced query set $\boldsymbol{P}' \in \mathbb{R}^{L \times 6}$, with a fixed sampling ratio $\alpha = L/N = 0.25$. This sampling strategy follows 3DShape2VecSet (Zhang et al., 2023) and serves to construct a compact query set for subsequent point cloud encoding.

To efficiently support weighted sampling with variable-length inputs, we modify the attention processor to enable fully batched attention without explicit for-loops, which significantly improves training efficiency.

### A.4. Network Architectures

We adopt the architecture of TripoSG (Li et al., 2025b) for the generation branch and initialize it with pretrained weights. For the reassembly branch, we extend the design of GARF (Li et al., 2025a) by incorporating query–key normalization (QK-norm) together with residual skip connections, which improves training stability when handling a large number of fragments.

Instead of pooling features from all tokens belonging to a part, we prepend a dedicated learnable token for each part. This token is explicitly used for flow-matching-based pose regression, allowing the network to decouple pose estimation from local geometric feature encoding.

*Table 7.* Network Architecture

| Hyperparameter | Assembly Branch | Generation Branch | Joint Adapter |
|---|---|---|---|
| Model Type | Custom DiT | TripoSG DiT (Li et al., 2025b) | Bi-dir Cross-Attn |
| Number of Layers | 21 | 21 | 21 |
| Hidden Dimension | 768 | 2048 | 512 |
| Attention Heads | 12 | 16 | 8 |
| Dim per Head | 64 | 128 | 64 |

Under the above settings, the original TripoSG (Li et al., 2025b) Transformer contains approximately 1.4B parameters. Our reassembly branch introduces an additional 216M parameters, while the joint adapter contributes 121M parameters.

We also experimented with a joint attention architecture that performs self-attention over concatenated tokens from different modalities and tasks, i.e., generation and reassembly. However, we found that such joint attention exhibits slower convergence and less stable training behavior in our setting while it cannot save parameters under same attention width and heads settings. Based on these observations, we stick with the bidirectional cross-attention design throughout all experiments.

### A.5. Training Details

We adopt a two-stage training strategy to stabilize optimization and effectively leverage the pretrained generation prior.

**Stage I: Assembly Warm-up.**  Instead of using the standard logit-normal sampling for training timesteps, we employ a **uniform sampling strategy** $t \sim \mathcal{U}(0, 1)$. In the reassembling task, the final adjustment of the pose of the parts requires a high-precision geometric alignment, which corresponds to the noise distribution near $t = 0$. Uniform sampling ensures that this tail distribution is sampled with sufficient frequency, preventing the network from neglecting fine-grained spatial adjustments that are critical for tight part mating.

**Stage II: Joint Training.**  For the second stage, we initialize the final linear projection layers of all joint adapters with **zero weights and biases**. This zero-initialization strategy ensures that the cross-branch interaction begins at zero magnitude, effectively preserving the pre-learned priors of the warmed-up assembly branch and the pretrained generation branch at the start of optimization.

Furthermore, we modify the classifier-free guidance (CFG) training setup. While the original TripoSG employs a conditioning dropout rate of $p = 0.1$, we significantly increase this rate to $p = 0.5$ during the training process. By exposing the model to the unconditional distribution more frequently, we encourage the network to learn a more robust distinction between conditional and unconditional scores. This allows us to apply a stronger guidance scale during inference, **forcing the generation branch to strictly adhere to the structural constraints provided by the assembly branch** rather than relying solely on its internal generative priors.

**Optimization.**  All experiments are optimized using AdamW with decoupled weight decay. We apply a linear learning rate warm-up at the beginning of training, followed by a cosine decay schedule. Unless otherwise specified, the same optimizer configuration is used for both training stages. Mixed-precision (bf16) training is used throughout to reduce memory consumption and improve training efficiency.

# B. Additional Experimental Setup

## B.1. FractureBEM Implementation Details

We employ the physically based fracture simulation method FractureBEM (Hahn & Wojtan, 2016) to generate realistic fracture patterns and curate the MorphSource (Boyer et al., 2016) dataset. For each fracture simulation, we define two opposing force application *regions* on the input shape to mimic compressive stresses experienced during fossilization. Each region is specified as a half-space induced by a plane orthogonal to a randomly sampled direction.

**Force application regions.** Let $\mathcal{V} = \{v_i \in \mathbb{R}^3\}_{i=1}^N$ denote the set of vertices of the input mesh. For each simulation trial, we first sample a unit direction vector $d$ uniformly at random on the unit sphere. Each vertex is projected onto the selected direction $d$ via the dot product

$$p_i = v_i^\top d, \qquad i = 1, \ldots, N. \tag{9}$$

Let

$$p_{\min} = \min_i p_i, \qquad p_{\max} = \max_i p_i, \tag{10}$$

and define the projection range as

$$\Delta p = p_{\max} - p_{\min}. \tag{11}$$

We define two disjoint vertex regions corresponding to the extreme projections along $d$. Specifically, using a fixed percentile factor $\alpha = 0.2$, we compute the thresholds

$$\tau_{\text{low}} = p_{\min} + \alpha \Delta p, \qquad \tau_{\text{high}} = p_{\max} - \alpha \Delta p. \tag{12}$$

The two force application regions are then defined as

$$\mathcal{R}_1 = \left\{ v_i \in \mathcal{V} \mid v_i^\top d < \tau_{\text{low}} \right\}, \tag{13}$$

$$\mathcal{R}_2 = \left\{ v_i \in \mathcal{V} \mid v_i^\top d > \tau_{\text{high}} \right\}. \tag{14}$$

These regions correspond to vertices located near the two extremes of the shape along the selected direction.

**Force directions.** In each simulation, the region $\mathcal{R}_1$ is treated as fixed, while a compressive load is applied to the opposing region $\mathcal{R}_2$. We apply the force to $\mathcal{R}_2$ along the direction $-d$, which pushes the selected region inward toward the object interior. To introduce variability and better approximate natural fossilization pressures, the force direction is further perturbed by a random angular jitter. Specifically, $f$ is rotated by a random angle sampled uniformly within a cone of $30°$ around the base direction $-d$.

**Fracture simulation.** The defined regions $\mathcal{R}_1$ and $\mathcal{R}_2$, together with their corresponding force directions, are then passed to `FractureBEM` to perform the fracture simulation under opposing compressive loads. Repeating this process with different random directions produces a diverse set of fracture patterns for dataset generation.

All input meshes are uniformly rescaled to a common canonical size prior to fracture simulation, such that the resolution of fracture surface details is comparable across the dataset. In addition, a fixed set of material parameters is used for all simulations. These choices ensure robustness and consistency across simulations.

## C. Additional Qualitative Results

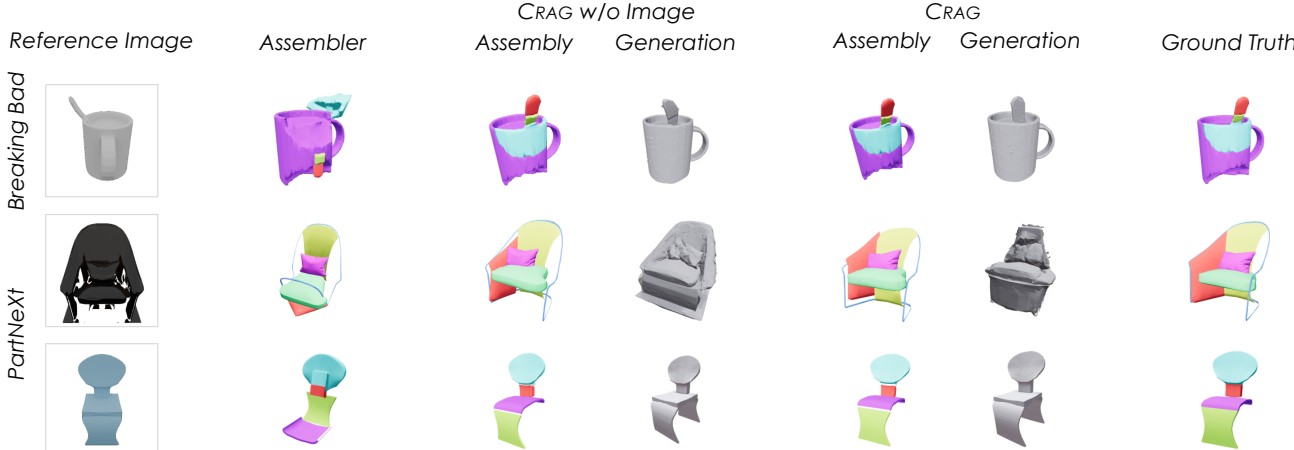

*Figure 8.* Qualitative results on PartNeXt (Wang et al., 2025a) and Breaking Bad (Sellán et al., 2022) We compare Assembler (Zhao et al., 2025a), CRAG without reference images, and CRAG given a reference image. CRAG simultaneously assembles the observed parts and synthesizes a plausible complete shape, and reference images further improve fidelity when available.

