# OpenReview forum: "CRAG: Can 3D Generative Models Help 3D Assembly?"
_ICML.cc/2026/Conference — ICML 2026 regular_

### Official Review · Reviewer_SGgh · 2026-03-09

**Soundness:** 3
**Presentation:** 3
**Significance:** 3
**Originality:** 3
**Overall Recommendation:** 5
**Confidence:** 5

**Summary:**

Traditional 3D part assembly methods typically treat the problem as a pure pose estimation task, focusing on aligning observed fragments through rigid transformations. CRAG is a unified framework that simultaneously performs 3D assembly and shape synthesis, addresssing 3D part assembly by arguing that 3D assembly should be coupled with global shape generation. By "imagining" the complete object, the model can better guide the placement of individual parts. TripoSG-VAE is employed for fragment embedding, while a Joint Flow Matching framework is utilized for assembly.

**Compliance With Llm Reviewing Policy:**

Affirmed.

**Final Justification:**

I really appreciate the authors' response, which has addressed all my concerns.

**Key Questions For Authors:**

Several shape completion methods are capable of generating the entire shape given only a single fragment. Comparisons with these methods should be included and discussed.

When some parts are missing, generation or completion approaches can also reconstruct the entire shape and may even outperform the proposed method. Therefore, the motivation of this work should be clarified more clearly.

In addition, it is unclear why the proposed method fails on thin fragments. Since the encoding framework does not rely on fractured surfaces and instead encodes the original surfaces, the reason for this limitation should be explained.

The robustness of the method to imperfect fragment assembly should also be investigated; for example, when at least one of the adjacent fragments is incomplete.

**Limitations:**

yes

**Strengths And Weaknesses:**

Strengths:

The core contribution of this paper is the transition of the 3D assembly task from a pure "geometric alignment" problem to a "generative inference" problem. This shift in perspective is highly forward-looking.

The authors claim to present a notable theme by establishing mutual constraints between the fragment pose space and the global shape latent space, achieving a co-evolution of assembly and generation. This design allows the model to dynamically "imagine" missing components during the assembly process.

Experiments demonstrate that even with missing parts, w/o reference images, CRAG maintains robust assembly performance.

Weaknesses:

With a reference image, the shape can be directly generated; therefore, the necessity of the two-branch framework should be discussed. Furthermore, several shape completion methods are capable of generating the entire shape given only a single fragment. Comparisons with these methods should be included and discussed. In addition, when some parts are missing, generation or completion approaches can also reconstruct the entire shape and may even outperform the proposed method. Therefore, the motivation of this work should be clarified more clearly.

A more detailed description of the architecture should be provided, for example, the specific design of the decoder.

The terminology should be unified throughout the paper, such as the use of fragment and part.

In addition, it is unclear why the proposed method fails on thin fragments. Since the encoding framework does not rely on fractured surfaces and instead encodes the original surfaces, the reason for this limitation should be explained.

The robustness of the proposed method to the size and number of fragments should be evaluated. In addition, the robustness of the method to imperfect fragment assembly should also be investigated; for example, when at least one of the adjacent fragments is incomplete.

---

> ### Author Rebuttal · Authors · 2026-03-31
>
> We thank the reviewer for recognizing CRAG's core contribution as a forward-looking shift from "geometric alignment" to "generative inference," and for acknowledging the co-evolution design and CRAG's robust performance under missing parts without reference images. Below we address the remaining comments.
>
> ---
>
> ## Q1&Q2&W1: Comparison with Shape Completion Methods
>
> We agree that shape completion is a relevant comparison. However, we note a key distinction in the **input setting**. Shape completion methods assume a single, pre-aligned partial shape, whereas our setting starts from **multiple unaligned fragments with arbitrary poses**, which completion methods cannot handle directly. Assembly is therefore indispensable as a prerequisite.
>
> To further address your concern, we compare the generation quality of CRAG with AdaPoinTr (Yu et al., ICCV 2021), a representative point cloud completion method, on PartNeXt (Missing-part setting). Notably, **AdaPoinTr receives fragments already placed at ground-truth poses** (i.e., a pre-assembled partial shape), whereas **CRAG starts from unassembled fragments and must simultaneously solve both assembly and generation**:
>
> |Method|Input Condition|CD-L1 (×10⁻²) ↓|CD-L2 (×10⁻³) ↓|F-Score@1% ↑|F-Score@5% ↑|EMD (×10⁻³) ↓|
> |---|---|---|---|---|---|---|
> |AdaPoinTr|GT-posed fragments|16.05|3.13|10.35%|51.81%|3.01|
> |CRAG w/o img|Unassembled fragments|11.98|1.84|12.88%|62.60%|2.06|
> |CRAG|Unassembled fragments + image|**10.07**|**1.43**|**16.09%**|**69.84%**|**1.78**|
>
> Despite operating under a strictly harder setting, CRAG outperforms AdaPoinTr across all generation metrics by a large margin.
>
> ---
>
> ## Q3&W4: Failure on Thin Fragments
>
> The failures on thin fragments stem from the **TSDF/SDF representation** used by TripoSG's VAE, not from our coupling mechanism. Specifically, when a fragment's thickness falls below the TSDF truncation distance, the signed distance fields from opposite surfaces overlap and cancel out, making the thin geometry unrepresentable in the latent space. Since both branches operate on this shared latent space, information lost at the encoding stage cannot be recovered by either branch. The same representation limitation also causes issues with slender components (cross-sections near voxel resolution) and non-watertight surfaces (where SDF's inside/outside distinction becomes ambiguous near open boundaries). This is a well-recognized limitation of field-based representations — recent work such as TRELLIS 2 (Xiang et al., 2025) explicitly introduces a field-free representation (O-Voxel) to handle such arbitrary topology. Replacing TripoSG with such topology-agnostic backbones is a promising future direction.
>
> ---
>
> ## Q4: Robustness to Incomplete Fragments
>
> An incomplete fragment is effectively equivalent to a missing-part scenario. We provide a systematic evaluation varying the missing ratio from 0% to 40% on PartNeXt, showing that CRAG's assembly quality (PA, RE, TE) remains remarkably stable across all ratios. Please refer to our response to Reviewer 5qLt (Q3) for the detailed table and analysis. Additionally, our experiments on MorphoSource and FRACTURA (Figures 3, 4, and 6) involve real-world bone fragments that are naturally eroded and incomplete, further demonstrating CRAG's robustness to imperfect fragment geometry in practice.
>
> ---
>
> ## W2: Detailed Description of the Architecture
>
> Thank you for this suggestion. Detailed architecture specifications (including the decoder design, layer configurations, and training hyperparameters) are provided in the Supplementary Material (Sec.A: Implementation Details). We will incorporate more architectural details into the main paper in the revised version.
>
> ---
>
> ## W3: Terminology
>
> Thank you for this suggestion. We will clarify this in the revision: we use part as the general term for an input piece, and fragment more specifically for fractured pieces arising from geometric breakage.
>
> ---
>
> ## W5: Evaluation on the Number of Fragments
>
> We report the part accuracy (PA) of CRAG on PartNeXt (Complete), compared with Assembler, grouped by the number of parts per shape:
>
> |PA / # Parts|2|3|4|5|6|7|8|9|10|
> |---|---|---|---|---|---|---|---|---|---|
> |CRAG (Ours)|95.0|91.2|84.7|70.9|65.6|62.0|57.0|58.9|53.9|
> |Assembler|83.6|76.0|64.7|54.0|44.5|38.8|33.6|34.0|31.1|
>
> Performance decreases gradually as the number of fragments increases, which is expected — more parts introduce greater combinatorial ambiguity.

---

> > ### Author Rebuttal · Reviewer_SGgh · 2026-04-01
> >
> > I appreciate the authors’ response, which provides a clear explanation of the distinction between the completion task and the assembly task. However, I am still unclear whether, with a reference image, the shape can be generated directly; therefore, the necessity of the two-branch framework should be discussed. Furthermore, with a reference image and only one fragment, the entire shape can also be generated, as some prior work has demonstrated. Therefore, the motivation for this work should be clarified more clearly.

---

> > > ### Author Response · Authors · 2026-04-03
> > >
> > > ## Follow-up Q1: Necessity of the Two-Branch Framework When a Reference Image Is Available
> > >
> > > We appreciate this follow-up question. To quantify how well generation performs with only a reference image, we provide the following comparison on the PartNeXt evaluation set (4,626 shapes):
> > >
> > > |Method|CD-L1 (×10⁻²) ↓|CD-L2 (×10⁻³) ↓|F-Score@1% ↑|F-Score@5% ↑|EMD (×10⁻³) ↓|Failure Rate ↓|
> > > |---|---|---|---|---|---|---|
> > > |TripoSG (image only)|9.12|11.23|14.63%|73.18%|16.67|**25.0%** (1157/4626)|
> > > |CRAG w/o img|9.26|11.46|15.02%|72.61%|16.96|0%|
> > > |CRAG|**6.53**|**5.83**|**20.49%**|**83.87%**|**12.92**|**0%**|
> > >
> > > From the **generation** perspective alone, our results suggest that **the two-branch design allows fragment evidence to provide structural constraints that a single reference image alone cannot reliably capture**. TripoSG fails entirely on 25% of the evaluation set (1,157/4,626 shapes) and achieves lower generation quality than CRAG across all metrics (e.g., CD-L2: 11.23 vs. 5.83, F-Score@5%: 73.18% vs. 83.87%). This is because **a single reference image often contains occlusion, ambiguity, and limited geometric detail** — fragment-level 3D evidence from the assembly branch provides structural constraints that the image alone cannot. Meanwhile, our two-branch framework enables the shape generation only from unaligned fragments without reference image (refer to CRAG w/o img in the table), which can’t be achieved by existing method.
> > >
> > > More importantly, we want to emphasize that **CRAG is not proposed for pure shape generation, but for 3D assembly**. As stated in the paper, the 3D assembly problem **takes a set of parts or fractured fragments as its essential input** and aims to recover the correct placement of these fragments while maintaining global shape coherence. This distinction is crucial: an image-to-3D generator may produce a plausible complete shape, but it does not tell where each fragment should be placed. The two-branch framework is therefore necessary **not only for improving generation quality and robustness**, but more fundamentally because **3D assembly requires both per-fragment pose prediction and holistic shape reasoning**. The assembly branch recovers fragment poses, while the generation branch provides global shape context. The two branches are thus complementary rather than redundant.
> > >
> > > ---
> > >
> > > ## Follow-up Q2: Comparison with Image + Single-Fragment Shape Generation Methods
> > >
> > > We thank the reviewer for pointing us to this relevant line of work. We conduct additional comparisons with two representative methods that accept both a reference image and a single fragment as input: SDFusion (Cheng et al., CVPR 2023) and MGPC (Liu et al., 2026). All methods are evaluated on *the PartNeXt Missing-part setting from 2-20 parts*. Notably, SDFusion and MGPC receive a **pre-aligned single fragment + reference image**, whereas CRAG starts from **multiple unassembled fragments**:
> > >
> > > |Method|CD-L1 (×10⁻²) ↓|CD-L2 (×10⁻³) ↓|F-Score@1% ↑|F-Score@5% ↑|EMD (×10⁻³) ↓|
> > > |---|---|---|---|---|---|
> > > |SDFusion (CVPR 2023)|22.51|23.93|1.90%|28.58%|28.60|
> > > |MGPC (arXiv 2026)|12.59|11.54|12.75%|70.54%|16.99|
> > > |CRAG w/o img|12.31|10.32|13.84%|74.11%|16.12|
> > > |CRAG|**9.39**|**5.76**|**18.23%**|**82.82%**|**13.09**|
> > >
> > > We highlight two key findings:
> > >
> > > **(1) CRAG significantly outperforms dedicated shape completion methods despite solving a harder problem.** CRAG with image achieves 82.82% F-Score@5%, compared to 70.54% for MGPC and 28.58% for SDFusion — even though CRAG starts from unassembled fragments while these methods receive a pre-aligned fragment. This demonstrates that jointly reasoning about assembly and generation produces better performance than treating completion as an isolated task.
> > >
> > > **(2) Even without any reference image, CRAG outperforms image-guided completion methods.** CRAG w/o img (no image, multiple unassembled fragments) achieves 74.11% F-Score@5% and 10.32 CD-L2, surpassing MGPC (70.54%, 11.54) which has access to both an image and a pre-aligned fragment.
> > >
> > > **Clarification on motivation.** We would like to respectfully re-emphasize that the motivation for CRAG is not to propose a better shape completion method, but to address a fundamentally different problem: **given N pose-unaligned 3D fragments (and an optional reference image), simultaneously recover the SE(3) pose for each fragment and synthesize the complete shape.** Shape completion methods like SDFusion and MGPC assume a single pre-aligned partial input and focus solely on hallucinating missing geometry. They provide no mechanism for assembling multiple fragments. CRAG’s two-branch design is a direct consequence of this problem formulation — the assembly branch predicts per-fragment poses, the generation branch synthesizes the holistic shape, and the Joint Adapter enables bidirectional information exchange so that each task benefits from the other.

---

### Official Review · Reviewer_5qLt · 2026-03-10

**Soundness:** 3
**Presentation:** 2
**Significance:** 2
**Originality:** 3
**Overall Recommendation:** 4
**Confidence:** 3

**Summary:**

The paper proposes CRAG, a unified framework that couples 3D assembly with whole-shape generation. The model contains two interacting branches: an assembly branch that estimates rigid transformations for each fragment using SE(3) flow matching, and a generation branch that synthesizes the complete object shape. A joint adapter connects the two branches and allows bidirectional information exchange so that global shape hypotheses can guide part placement, while assembled parts provide structural evidence for generation. The model is built on top of a pretrained generative backbone and trained with a two-stage strategy that first learns assembly and then jointly optimizes both tasks. Experiments on several datasets show improved assembly accuracy and better reconstruction of complete shapes, especially in scenarios with ambiguous alignment or missing fragments.

**Compliance With Llm Reviewing Policy:**

Affirmed.

**Final Justification:**

Based on the above discussion, I tend to choose "weak accept".

**Key Questions For Authors:**

1. Contribution of the coupling mechanism: How much of the improvement comes from the joint assembly–generation coupling versus simply using the pretrained generative model as a shape prior? A clearer ablation isolating the role of the joint adapter would help clarify this.
2. Dependence on the generative backbone: How sensitive is the method to the choice or quality of the pretrained generator (e.g., TripoSG)? Would the approach still work if the generator produces lower-quality shapes or if the domain differs from the training data?

3. Missing-fragment scenarios: The paper claims robustness to missing pieces. Could the authors provide a more systematic evaluation where the percentage of missing parts is varied to show how performance changes?

**Limitations:**

yes

**Strengths And Weaknesses:**

Strengths:
1. The core idea of coupling assembly and generation is intuitive and well motivated. Using a global shape prior from generative models to regularize fragment alignment addresses a known weakness of pose-only approaches.

2. The architecture is conceptually clean. The two-branch design with a shared adapter allows information to flow in both directions, which makes sense from a reasoning perspective: assembly benefits from global context while generation benefits from part-level evidence.

3. The experimental evaluation is relatively thorough. The paper evaluates on multiple datasets and compares with several recent baselines. The results suggest that the proposed coupling strategy improves both alignment quality and shape completion, particularly when parts are missing.

Weaknesses:
1. While the idea of combining assembly with generative modeling is interesting, the level of technical novelty in the architecture itself appears somewhat incremental. The method relies heavily on an existing generative backbone and flow-matching formulations, and the main contribution lies in how these components are combined.

2. The paper could better clarify which component contributes most to the improvements. Since the system includes a pretrained generator, flow-based pose estimation, and the joint adapter, it is somewhat difficult to determine whether the gains mainly come from the generative prior or from the proposed coupling mechanism.

3.  Some experimental details are relatively limited. For example, the training setup, computational cost, and inference time compared to existing assembly methods are not discussed in depth.

---

> ### Author Rebuttal · Authors · 2026-03-31
>
> We sincerely thank the reviewer for recognizing **the intuitive motivation of our formulation**, **the clarity of our bidirectional two-branch design**, and **the thoroughness of our experiments**.
>
> ---
>
> ## Q1&W2: Contribution of the Coupling Mechanism.
>
> We initially explored feeding a reference image to the frozen pretrained generative model as a shape prior for assembly without the joint adapter. This trivial design does not work — it even performs worse than GARF.
>
> Crucially, such a one-way design cannot allow observed fragments to improve generation (refer to our response to Reviewer YRf7, Q1). Our bidirectional coupling enables a new capability: assembly evidence improves generation, especially under missing-part settings (e.g., Fig. 4). Furthermore, even without a reference image, CRAG generates plausible complete shapes from fragment evidence alone and improves assembly over GARF/RPF (Table 1), confirming that the benefit comes from the coupled interaction itself, not from a one-way prior.
>
> ---
>
> ## Q2: Sensitivity to the Generative Backbone and Domain Shift.
>
> We address it from two angles: (1) robustness to domain shift, and (2) architectural design choices that reduce dependence on the pretrained generator.
>
> **(1) CRAG generalizes well across domains far from the generative prior’s training data.** TripoSG’s VAE is pretrained on Objaverse, a large-scale dataset of synthetic everyday objects. However, our evaluation spans multiple domains with increasing distance from this training distribution:
>
> |Dataset|Domain|Relation to Objaverse|PA ↑ (Complete)|PA ↑ (Missing)|
> |---|---|---|---|---|
> |PartNeXt|Semantic parts of everyday objects|Closest|65.89|71.81|
> |Breaking Bad|Fracture reassembly of everyday objects|Moderate shift (fracture geometry)|94.68|92.07|
>
> On Breaking Bad, which involves fracture surfaces that differ substantially from the clean semantic boundaries in Objaverse, CRAG still achieves strong performance (PA: 94.68, CD: 0.21 in the complete setting). On MorphoSource and FRACTURA — domains entirely unseen during TripoSG’s pretraining — CRAG produces coherent assemblies and plausible shape completions as shown in Figs. 3, 4, and 6. These results demonstrate that the learned generative prior transfers effectively across domains.
>
> **(2) Our architectural design mitigates over-reliance on the pretrained generator**:
>
> - **Two-stage training**. In Stage 1, the assembly branch is trained independently for 100k steps, learning robust pose estimation without any dependence on the generation branch. In Stage 2, the generation branch is activated and jointly finetuned for 150k steps. This ensures the assembly branch does not collapse onto a potentially imperfect generative prior.
> - **Zero-initialized Joint Adapter**. The adapter’s output projection layers are initialized to zero, so the coupling starts as an identity mapping. The generative prior is incorporated gradually during training, allowing the model to learn how much to rely on the prior rather than being forced to depend on it.
>
> ---
>
> ## Q3: Systematic Evaluation with Varying Missing-Part Ratios.
>
> We provide a systematic evaluation on PartNeXt (2–20 parts per shape) where we vary the percentage of randomly dropped fragments from 0% to 40%. Results are reported for *CRAG* *w Image*:
>
> |Missing Ratio|RE ↓|TE ↓|PA ↑|CD ↓|
> |---|---|---|---|---|
> |0% (Complete)|45.12|9.13|65.89|2.40|
> |10%|45.72|9.09|65.72|2.57|
> |20%|44.74|9.08|65.99|3.06|
> |30%|43.66|9.07|66.03|3.71|
> |40%|42.60|9.09|65.92|4.70|
>
> **Assembly quality remains remarkably stable.** PA stays within a narrow range of 65.72–66.03 across all missing ratios, and RE/TE show virtually no degradation. This demonstrates that the generative prior effectively compensates for the loss of fragment evidence, maintaining robust pose estimation even when up to 40% of parts are missing.
>
> ---
>
> ## W1: Technical Novelty.
>
> Our technical novelty lies not in redesigning the standalone assembly backbone or 3D generator, but in **the synergy of their unification, enabling capabilities neither can achieve alone**. This is broadly consistent with recent unified multimodal models (e.g., MetaMorph and BLIP3-o), whose impact comes from organizing pretrained components into a joint system rather than redesigning underlying modules.
>
> ---
>
> ## W3: Training Setup, Computational Cost and Inference Performance.
>
> The full training process took approximately 60 hours using 32 H200 GPUs. For evaluation, we compared our method with GARF, with both approaches using 50 denoising steps. It is important to note that we report the inference time excluding the shape decoding process, as shape decoding accounts for a significant portion of the overall inference time. The inference time was tested on a single H200 GPU.
>
> |Method|Inference Time (s)|
> |---|---|
> |GARF|0.41|
> |CRAG w.o Shape Decoding|49.47|
> |CRAG|315.75|

---

> > ### Author Rebuttal · Reviewer_5qLt · 2026-04-03
> >
> > Thank you for your reply. My concerns have been addressed.

---

### Official Review · Reviewer_YRf7 · 2026-03-13

**Soundness:** 2
**Presentation:** 3
**Significance:** 3
**Originality:** 2
**Overall Recommendation:** 4
**Confidence:** 4

**Summary:**

This paper proposes CRAG, a unified framework that couples 3D assembly and whole-shape generation. The key idea is to move beyond treating assembly as pure pose estimation and instead jointly infer fragment poses and a plausible complete shape. The method reuses the TripoSG VAE as a shared latent space, employs two parallel flow-matching branches for assembly and generation, and introduces a Joint Adapter with bidirectional cross-attention to exchange information between the two tasks. Experiments on PartNeXt, Breaking Bad, MorphoSource, and qualitative evaluation on real fracture data suggest that the approach improves assembly accuracy and remains robust when parts are missing.

**Compliance With Llm Reviewing Policy:**

Affirmed.

**Final Justification:**

My concerns have been addressed. Therefore, I would like to maintain my score as a week accept.

**Key Questions For Authors:**

Can the authors provide stronger quantitative evidence that part-level evidence improves generation, beyond the qualitative examples in Figure 5

**Strengths And Weaknesses:**

Strengths:

1. Reformulating 3D assembly as a coupled assembly-and-generation task is intuitive and meaningful, especially in realistic settings where parts may be missing or heavily degraded.

2. The shared VAE latent space, the dual-branch flow-matching design, and the Joint Adapter together form a clear framework rather than a loose collection of components. The architecture in Figure 2 communicates the approach effectively.

3. The paper evaluates complete and missing-part settings, compares image-free and image-conditioned variants, includes both semantic-part assembly and fracture reassembly, and provides ablations and failure cases.

Weaknesses:

1. The main contribution lies in coupling existing ingredients—a pretrained generative prior, an assembly flow branch, and bidirectional interaction through adapters—rather than introducing a fundamentally new modeling primitive. The formulation is interesting, but the core technical mechanism itself feels more integrative than deeply novel.

2. The “mutual reinforcement” claim is only partially validated. The evidence that generation helps assembly is fairly strong, especially in the missing-part setting. However, the reverse direction—that part-level evidence substantially improves generation—is supported much more weakly, mostly through qualitative examples and cautious wording such as “to some extent.” This makes the central bidirectional claim feel somewhat asymmetric.

3. The failure analysis is helpful but still limited. Figure 7 identifies several meaningful failure modes, including thin-shell ambiguities and TSDF/VAE issues with slender or non-watertight shapes, but the paper does not deeply analyze whether these failures stem from the representation, the generative prior, or the coupling mechanism itself.

---

> ### Author Rebuttal · Authors · 2026-03-30
>
> We are deeply encouraged by your recognition of the intuitive value of our coupled assembly-and-generation formulation, the clear architecture of our unified framework, and the comprehensiveness of our evaluations.
>
> ---
>
> ## Q1&W2: Quantitative Evidence for Part-level Assembly Improving Generation.
>
> We provide quantitative generation evaluation on the PartNeXt evaluation set (4,626 shapes) for the assembly→generation direction, comparing three setups:
>
> - **TripoSG**: image-conditioned generation only (no assembly information)
> - **CRAG w/o img**: generation guided by part-level assembly evidence (no reference image)
> - **CRAG**: full model with both image conditioning and assembly coupling
>
> |Method|CD-L1 (×10⁻²) ↓|CD-L2 (×10⁻³) ↓|F-Score@1% ↑|F-Score@5% ↑|EMD (×10⁻³) ↓|Failure Rate ↓|
> |---|---|---|---|---|---|---|
> |TripoSG|9.12|11.23|14.63%|73.18%|16.67|**25.0%** (1157/4626)|
> |CRAG w/o img|9.26|11.46|15.02%|72.61%|16.96|0%|
> |CRAG|**6.53**|**5.83**|**20.49%**|**83.87%**|**12.92**|**0%**|
>
> We highlight two key findings:
>
> **(1) Part-level evidence significantly improves image-conditioned generation (TripoSG → CRAG).** With the reference image, adding assembly coupling yields gains across all metrics: CD-L2 drops by 48.1% (11.23 → 5.83) and F-Score@1% improves by 40.1% (14.63% → 20.49%), demonstrating that fragment geometry provides structural constraints that complement image cues for resolving occluded or ambiguous regions.
>
> **(2) Part-level evidence alone achieves comparable generation to image-conditioned TripoSG — without failing.** CRAG w/o img relies entirely on fragment geometry with no reference image, yet achieves comparable generation quality to TripoSG (CD-L1: 9.26 vs. 9.12), while TripoSG fails on 25% of the evaluation set (1,157/4,626 shapes).
>
> These results provide quantitative support for the bidirectional nature of our framework: generation helps assembly (as shown in Table 1 of the paper), and assembly helps generation (as shown above). We will add this table to the revised paper accordingly.
>
> ---
>
> ## W1: Core Technical Mechanism.
>
> We respectfully clarify that our novelty is not a new standalone modeling primitive, but **a nontrivial coupling mechanism** that organizes assembly and generation into a joint system **with capabilities neither achieves alone**. This is broadly consistent with recent unified multimodal models (e.g., MetaMorph and BLIP3-o), whose impact comes from organizing pretrained components into a joint system rather than redesigning underlying modules. CRAG brings a new insight: 3D assembly and generation are mutually reinforcing tasks that benefit from joint optimization.
>
> Our experiments show that **trivially combining assembly and generation does not work**. Even when fragments are placed at GT poses, a shape completion method (AdaPoinTr) reconstructs worse than CRAG (F-Score@5%: 51.81% vs. 69.84%). Conversely, injecting GT shape latents as a static condition into the assembly does not yield gains. In contrast, CRAG performs well even in the image-free setting (CRAG w/o img, Table 1 and Figs. 3–4). Without a reference image, the generation must rely on part-level features from the assembly, yet it still synthesizes globally consistent shapes. Our design addresses the representational gap between variable-length fragment observations and fixed-size shape latents, and enables iterative mutual refinement between local fragment evidence and global shape hypotheses, **rather than a trivial combination identifiable a priori**.
>
> ---
>
> ## W3: Deeper Analysis of Failure Modes.
>
> **All three failure modes stem from the representation (TSDF/SDF), not from the coupling mechanism.**
>
> **(1) Thin-shell structures.** when thickness falls below the TSDF truncation distance, SDFs from opposite surfaces cancel out, making the geometry unrepresentable in the latent space — information lost at encoding cannot be recovered by either branch.
>
> **(2) Slender components.** cross-sections near voxel resolution are underrepresented in the discrete grid and can’t capture fine elongated structures, leading to broken geometry.
>
> **(3) Non-watertight surfaces.** SDF requires a well-defined inside/outside distinction; for open boundaries, ambiguous signed distances cause hole-closing and sheet-thickening artifacts that propagate through the coupling.
>
> This is a widely recognized limitation of SDF-based representations — TRELLIS 2 (Xiang et al., 2025) explicitly introduces a field-free representation (O-Voxel) to handle arbitrary topology. Replacing TripoSG with such topology-agnostic backbones is a promising future direction.

---

> > ### Author Rebuttal · Reviewer_YRf7 · 2026-04-02
> >
> > The additional experiments and analysis have addressed my concerns. I tend to maintain my score.

---

### Official Review · Reviewer_Rdn1 · 2026-03-23

**Soundness:** 2
**Presentation:** 2
**Significance:** 2
**Originality:** 2
**Overall Recommendation:** 4
**Confidence:** 3

**Summary:**

CRAG (Coupled ReAssembly and Generation) is a unified 3D assembly framework that integrates fragment pose estimation with holistic shape generation. Moving beyond traditional methods that treat assembly as a pure rigid transformation problem, CRAG simulates human reasoning by using a global shape prior to resolve local alignment ambiguities. It utilizes a Mixture-of-Transformers with two parallel branches—an Assembly Branch for predicting SE(3) flows and a Generation Branch for synthesizing shape latents.  It proposes a Joint Adapter with a bi-directional attention mechanism that bridges the branches, allowing fragment details to inform the global shape while the imagined whole guides fragment alignment. The experimental results show that CRAG seems to robust to missing parts, capable of "hallucinating" absent geometry to synthesize a complete, plausible shape while accurately positioning observed fragments.

**Compliance With Llm Reviewing Policy:**

Affirmed.

**Final Justification:**

Although the current implementation is mainly a combination of existing components, the problem addressed represents a novel exploratory direction. For this reason, I would incline toward a weak accept. I encourage the authors to further improve and refine the paper.

**Key Questions For Authors:**

1. Given that the framework relies on existing components like GARF and TripoSG, can you clarify the principled structural or geometric advantages of your "loose coupling" via cross-attention compared to a more integrated architectural approach?

2. Since CRAG utilizes a significantly deeper assembly branch (21 layers) and a large generative backbone, how much of the performance gain is truly due to the "coupled" design rather than simply the substantial increase in model capacity and parameters?

3. The ablation study suggests the 3D generative prior provides only marginal gains; could you provide more evidence or specific scenarios where the generative branch is indispensable for successful assembly?

4. Why was a holistic (global) shape prior chosen over a part-level generative approach (e.g., PartCrafter), which might naturally align better with the compositional and structured nature of 3D assembly?

**Limitations:**

1. Limited Technical Novelty: The framework largely aggregates existing architectural designs, such as GARF-style assembly and simple cross-attention mechanisms, lacking a fundamental breakthrough in how geometry and structure are integrated.

2. High Computational Cost: The improved results come at the expense of a significantly larger parameter count and model depth, making it difficult to isolate the effectiveness of the proposed methodology from the benefits of increased scale.

3. Representation Mismatch: There is an inherent disconnect between using a global 3D generative model and the task of structured part assembly, as the model may struggle to address fine-grained, part-level missing information.

4. Questionable Contribution of Priors: The actual impact of the 3D generative prior appears marginal in quantitative tests, raising concerns about whether the complexity of adding a generative branch is justified by the resulting performance.

**Strengths And Weaknesses:**

Strength:

This work propose a method for 3D assembly under missing parts. The author try to address incomplete part assembly remains limited in prior work. In addition, the paper introduces 3D generative priors into the assembly pipeline, which is an interesting and timely direction.

Weakness
1. The overall framework shows little technical innovation. It largely consists of existing components, with the assembly branch built upon prior designs (e.g., GARF-style architectures), and the interaction with the generative model implemented via relatively simple cross-attention. This results in a loose coupling that operates mainly at the feature level, rather than a principled integration of structure or geometry.

2.Although the paper reports improved performance, the model also introduces significantly more parameters (e.g., deeper assembly branch compared to prior methods such as GARF). As such, the improvements may be partially attributed to increased model capacity rather than the proposed design. Furthermore, the ablation study suggests that the 3D generative prior itself provides only marginal gains, raising questions about its actual contribution to the overall performance.

3. The paper adopts a global 3D generative model (e.g., TriPoSG-style), which is not well aligned with the inherently compositional nature of 3D assembly. In practice, missing information typically occurs at the part level, suggesting that part-level generation (e.g., recent works such as PartCrafter, 2025/06) would be a more natural and effective direction. The use of a holistic shape prior introduces a mismatch between representation (global shape generation) and task requirements (structured part assembly), and this design choice is not sufficiently justified.

---

> ### Author Rebuttal · Authors · 2026-03-31
>
> We appreciate your recognition that CRAG is an **interesting and timely direction**. We believe our response has addressed your concerns, and we would be grateful if you could take another look in light of the clarifications we have provided.
>
> ---
>
> ## Q1 & W1: Technical Innovation and A “Loose Coupling”.
>
> A1: Our technical novelty lies not in redesigning the standalone assembly backbone or 3D generator, but **in the synergy of their unification, enabling capabilities neither can achieve alone**. This is broadly consistent with recent unified multimodal models (e.g., MetaMorph and BLIP3-o), whose impact comes from organizing pretrained components into a joint system rather than redesigning underlying modules. CRAG brings a new insight: 3D assembly and generation are mutually reinforcing tasks that benefit from joint optimization. **Reviewers YRf7, 5qLt, and SGgh** also recognized that CRAG forms “**a clear framework rather than a loose collection of components**,” with “**conceptually clean**” architecture and “**mutual constraints**”.
>
> Our experiments show that **trivially combining assembly and generation does not work**. Even when fragments are placed at GT poses, a shape completion method (AdaPoinTr) reconstructs worse than CRAG (F-Score@5%: 51.81% vs. 69.84%). Conversely, injecting GT shape latents as a static condition into the assembly does not yield gains. In contrast, CRAG performs well even in the image-free setting (CRAG w/o img, Table 1 and Figs. 3–4). Without a reference image, the generation must rely on part-level features from the assembly, yet it still synthesizes globally consistent shapes. Our design thus enables progressive refinement of fragment evidence and shape hypotheses, **rather than a trivial combination identifiable a priori**.
>
> ---
>
> ## Q2 & W2: Performance Gain Due to the "Coupled" Design or More Parameters.
>
> We scale the original GARF architecture from 6 layers to 21 layers (matching CRAG’s assembly branch depth) while keeping everything else identical. The updated ablation results on PartNeXt are shown below:
>
> |Setup|Enc.|Branch|Assembly Layers|RE ↓|TE ↓|PA ↑|CD ↓|
> |---|---|---|---|---|---|---|---|
> |① (GARF)|PTv3|A|6|52.52|10.68|58.19|3.10|
> |② (GARF)|PTv3|A|21|53.40|12.83|56.15|4.27|
> |③|VAE|A|21|47.16|10.12|60.01|3.61|
> |④ (CRAG)|VAE|A + G|21|**45.12**|**9.13**|**65.89**|**2.40**|
>
> We highlight three key comparisons that isolate each source of improvement:
>
> **(1) Increased capacity alone hurts (① → ②).**
>
> **(2) Shared VAE representation is a key enabler (② → ③).**
>
> **(3) The coupled generation branch provides further gains (③ → ④).**
>
> ---
>
> ## Q3: Marginal Gains of the 3D Generative Prior.
>
> We respectfully clarify that our claim is that generation helps assembly, rather than being “indispensable”; more importantly, its contribution is clearly not marginal. We present ablation results comparing CRAG with and without the generation branch under both Complete and Missing-part settings on PartNeXt:
>
> **Complete (all parts observed):**
>
> |Setup|RE ↓|TE ↓|PA ↑|CD ↓|
> |---|---|---|---|---|
> |w/o Generation|47.16|10.12|60.01|3.61|
> |CRAG|**45.12**|**9.13**|**65.89**|**2.40**|
> |Δ|−2.04|−0.99|+5.88|−1.21|
>
> **Missing (with missing parts):**
>
> |Setup|RE ↓|TE ↓|PA ↑|CD ↓|
> |---|---|---|---|---|
> |w/o Generation|**40.72**|8.02|68.23|5.13|
> |CRAG|42.33|**7.86**|**71.81**|**4.21**|
> |Δ|+1.61|−0.16|+3.58|−0.92|
>
> In the Complete setting, the generation branch alone accounts for 5.88 of the 21.71 PA gain over Assembler (27%). It reduces CD from 3.61 to 2.40 in the CRAG vs. w/o Generation ablation. For reference, PMTR (ICML 2024) improved pairwise CD from 0.51 to 0.25 over the previous best method, so our gains are clearly not marginal.
>
> Our contribution is not only numerical improvement, but a new capability absent from prior assembly methods.
>
> ---
>
> ## Q4 & W3: A Holistic Shape Prior or A Part-Level Generative Approach (e.g., PartCrafter)?
>
> A4: We carefully considered part-level generation (e.g., PartCrafter) but chose a holistic prior for three reasons:
> **(1)** **The Unknown Part Count**: PartCrafter strictly requires the exact number of parts (N) as a input. But the number of missing fragments is unknown. A global prior naturally bypasses this by imaging the holistic structure without needing a predefined part count.
> **(2)** **Inability to Condition on Observed 3D Geometry**: PartCrafter is an image-to-parts model whose output geometry is purely determined by the 2D image. It cannot accept 3D fragments as geometric conditions, whereas CRAG strictly conditions generation on observed fragments via cross-attention.
> **(3)** **Computational and Decoding Costs**: PartCrafter fine-tunes TripoSG with 1,024 tokens per part (10,240 tokens for 10 parts) and requires separate decoding per part. CRAG jointly processes all fragments with a fixed token budget (5,000 tokens total) and decodes only a single global latent (2,048 tokens), making it more efficient.

---

### Decision · Program_Chairs · 2026-04-30

**Decision:**

Accept (regular)

**Comment:**

After rebuttal and discussion, all the reviewers proposed to accept this submission, including Reviewer Rdn1 who originally proposed rejection and did not acknowledge the rebuttal formally.

The paper proposes a novel approach to 3D assembly that seems to be a good solution to the case where parts are missing.
The AC believes this is interesting, the paper is overall well written, and the method and experiments are technically sound--especially with the complements brought by the authors in the rebuttals.